# Dimensions of Athlete-Coach Relationship and Sport Anxiety as Predictors of the Changes in Psychomotor and Motivational Welfare of Child Athletes after the Implementation of the Psychological Workshops for Coaches

**DOI:** 10.3390/ijerph19063462

**Published:** 2022-03-15

**Authors:** Dominika Wilczyńska, Tamara Walczak-Kozłowska, David Alarcón, Dominika Zakrzewska, Jose Carlos Jaenes

**Affiliations:** 1Physical Education and Social Sciences Department, Gdańsk Unversity of Physical Education and Sport, 80-336 Gdańsk, Poland; zakrzewska.dominika@yahoo.pl; 2Division of Neuropsychology, Institute of Psychology, Department of the Social Sciences, University of Gdańsk, 80-309 Gdańsk, Poland; tamara.walczak@ug.edu.pl; 3Social Anthropology, Basic Psychology and Public Health Department, Pablo de Olavide University, 41013 Seville, Spain; dalarub@upo.es (D.A.); jcjaesan@upo.es (J.C.J.)

**Keywords:** Athlete-Coach relationship, sport emotions and motives, young athletes, sport psychology

## Abstract

(1) Background: Coach workshops based on seven principles (inspiration, explanation, expectation, support, reward, appreciation, growth, and winning) enhance the sport experience of adult athletes. Here, we report effects of such workshops with coaches of child athletes and the predictors of those changes. (2) Methods: Study participants were 8 coaches of 57 children aged between 9 and 12 years old (girls practicing gymnastics and boys practicing football). Three coaches of 28 children attended three workshops over 12 weeks, while a control group of 5 coaches of 29 children attended no workshops. Measures of motivation, relationships, anxiety, and psychomotor performance were taken on children before and after the intervention. (3) Results: There were significant effects of the workshop on motivation and psychomotor performance. The analysis of the predictors the intervention used in this study might be effective for enhancing psychomotor performance and motivation while considering components of Athlete-Coach relationship and anxiety levels as moderators. (4) Conclusions: The beneficial effects of the workshop are encouraging but need to be investigated with higher numbers of coaches and children from various sport disciplines.

## 1. Introduction

There has been a body of studies underlying the significance of the psychological aspects in sports performance and competition [1,2,3,4,5]. Still, more studies need to focus on very young participants of sport (i.e., under ten years of age). The reason for this situation probably lies in a lack of validated instruments to measure selected psychological parameters, and children under the age of 12 are thought not to discriminate maturely between different psychological concepts [6]. Nonetheless, the need to know more about their experiences is essential. The success in adult and youth sport is the result of the interaction between many aspects, from the anthropometric, physiological characteristics of the body to the technical, coordinative abilities and mental condition [7,8,9,10,11]. Some researchers emphasize that such psychological factors as intrinsic motivation, social support, relations with coach and peers, sport enjoyment, perceived competence, task/mastery orientation, and effective learning development are crucial components of growth in youth sport in particular [12,13,14,15,16,17,18,19]. Sports participation should bring positive emotions to children and create positive associations and experiences to develop passion and perseverance [20,21]. However, experiencing anxiety is not rare in youth sport, and psychological literature reveals that anxiety is accompanied by different emotions, impacting cognitive processes and psychomotor performance. There have been studies showing opposite effects. On the one hand, presenting that anxiety can be helpful when performing a simple task [22]; on the other hand, convincing that anxiety can be alarming when dealing with a complicated task or in the process of decision-making or creative thinking [21,23]. However, the anxiety should be considered individually, as every athlete may have an “individual zone of optimal functioning” where they operate most effectively. Moreover, the interpretation of the symptoms of athletes’ anxiety is also essential. The level of an individual’s anxiety can have both positive and negative effects on the effectiveness of an action depending on the athletes’ confidence in dealing with it [24,25,26,27]. As anxiety is one of the essential factors in athlete optimal functioning, the authors of the current study considered it a predictor of young athletes’ motivation and psychomotor performance [22].

Motivation, as mentioned before, is one of the factors essential for the positive and healthy growth of young athletes. Sports motivation has been studied from different theoretical backgrounds over the years, and the most remarkable and comprehensive construct for understanding human behavior is Deci and Ryan’s self-determination theory [28], which considers motivation as a complex, multidimensional construct in which people are motivated for different reasons. The authors distinguished six types of behavioral regulations which can be placed along the continuum in which motivation is progressively internalized. Amotivation lies at the end of the continuum and refers to the lack of self-determination and the complete lack of motivational drive for any sports practice and activity. The other end of the continuum is intrinsic motivation, which is derived from internal sources, the pure pleasure and satisfaction of engaging in the behavior, and enjoyment derived from participating in sport. Between those two lie four types of extrinsic motivation: external and introjected, which represent regulation driven from the external locus of causality and are more controlling in nature; identified and integrated regulations represent the more autonomous types of extrinsic motivation. Ryan and Deci [29] suggest that autonomously motivated athletes will be characterized with positive and adaptive behaviors than more extrinsic-regulated athletes who could be more prone to burnout and dropout. Research shows that autonomous motivation is positively correlated with better performance and subjective wellbeing [30,31,32] in comparison to controlled types of motivation related to maladaptive outcomes, including dropout [33] and burnout [34]. The research based on self-determination theory [29] points out that young athletes’ feelings of autonomy, competence, and relatedness are essential and necessary to nourish in youth sport. If those three needs are satisfied, children and teenagers are more likely to experience wellbeing in sports. Deci and Ryan also emphasize that the social environment created by significant others, such as the coach, increases the likelihood of remaining in the sport. The autonomy-supportive and socially supportive coach behaviors are linked to more positive consequences for young sports participants than controlling coaches who can lead to detrimental consequences [35,36]. Therefore, the interventions which aim to educate coaches to create empowering motivational climates that tend to be marked with task-involving, autonomy, and socially supportive features to promote mentally and physically healthy and happy young athletes are worth investigating. That is the goal of the intervention implemented in the current study, which focuses on the youngest participants of sport, children.

In the present investigation, the authors used the i7W model of Poczwardowski et al. (2015) in the workshops for coaches to maximize selected psychological factors of children practicing sport [37]. This is the first published project which investigates the i7W model on child athletes [38,39]. The i7W model was justified by three critical theoretical frameworks: self-determination theory, coach–athlete relationship models, and transformational leadership. Furthermore, the model was supported by selected correlates on an individual level (flow, self-efficacy) and a team level (team cohesion). The results showed that athletes whose coaches practiced the i7W model in eight workshops (lasting one and half hours each) estimated their relationships with coaches to be significantly stronger and their self-confidence and group cohesion to be higher. Athletes especially perceived an increase in their coaches’ usage of behaviors from four categories: explain, reward, expect, and appreciate [40].

Two general aims guided the current study. The first aim was to investigate the impact of workshops for coaches on their young athletes’ autonomous motivation, psychomotor performance, and positive affect while performing in a sport context, and Athlete-Coach positive relationship. The second aim was to determine if such aspects as competitive anxiety and selected features of the Athlete-Coach relationship would predict the change in motivation and psychomotor performance.

## 2. Materials and Methods

### 2.1. Participants

Participants of the study were children at the age of nine to 12 years practicing football, and rhythmic and artistic gymnastics. Twenty-eight children from the experimental group practiced with three coaches (two coaches with 15 footballers; one coach with 13 gymnasts), while 29 children from the control group practiced with five coaches (four coaches with 19 footballers; one coach with ten gymnasts). The study began with 30 children in the experimental groups and 31 in the control groups. Four children were excluded from the study because they did not attend the laboratory for the post-test (two because of illness). Written informed consent from coaches and from parents or guardians of the children was a prerequisite to participate in the study. 

We chose gymnastics and football because of their early specialization and the exclusion of other sports, which means participants could not practice other sport disciplines. Malina (2010) and Waldron et al. (2020) define early specialization as participation in a single sport discipline at or before the age of 12, with a high volume of training [41,42]. In the current study, there were no significant differences between children from the control and intervention groups in age (10.3 ± 0.9 and 9.6 ± 1.1 years, respectively, mean ± SD), experience in their sport (4.4 ± 1.6 and 4.5 ± 1.5 years), experience with their current coach (3.1 ± 2.1 and 2.3 ± 1.6 years), and week training loads (4.9 ± 1.4 and 5.1 ± 1.6 training per week; 121 ± 43.5 and 131.7 ± 45.7 min per training). A Pearson’s chi-square test was performed to check for group differences by gender or discipline, finding that the experimental and control groups were balanced by gender or discipline (χ^2^ = 2.12, *p* = 0.14). Children were investigated in autumn and early wintertime, during the school semester, preparatory and competitive periods for children practicing football and gymnastics in Poland.

### 2.2. Design and Procedures

The workshops were based on the i7W model of Poczwardowski et al. (2015). The activities in this model focus on the coach–player relationship to promote psychosocial development and sports achievement [37]. The model is based on principles, which form the Polish abbreviation i7W. Those principles are “i” and seven “w”s: inspire (inspiruj), explain (wyjaśniaj), expect (wymagaj), support (wspieraj), reward (wnagradzaj), and appreciate (wyróżniaj), which are expected to have a positive effect on athletes, and thereby contribute to growth (wzrastać) and winning (wygrywać) [37]. The aim of the workshops was to increase the coaches’ ability to use the behavioral dimensions of the i7W model: inspire, support, explain, expect, reward, and appreciate.

There were three workshops, each lasting 6 h. The topics of the workshops are summarized in Table 1 (for more details contact the corresponding author).

Coaches were recruited by email contact with football and gymnastics clubs in the Pomeranian and Warmian-Masurian regions of Poland. Eventually, eight coaches from four clubs who expressed interest in participation were then sent an email with detailed information about the i7W workshop aa well as the date of the first meeting. The two coaches of gymnastics and six coaches of football were randomly assigned to the control and experimental groups separately (for the football coaches, randomization was stratified by club) after the initial meeting. The coaches in the experimental group were presented with more detailed information about the workshops, while those in the control group were told they would receive assessments of psychological and psychomotor characteristics of their athletes. After the final confirmation of the participation in the study, the coaches received information for parents and guardians of the children about the study’s aim, nature, and practice. The initial meeting with children took place in a research laboratory, where children were familiarized with all the tests required for the study. The implementation of the study was then divided into three phases. In the first phase, prior to the intervention (workshops), the intervention and control groups underwent psychological and psychomotor tests (listed below) in the laboratory. The second phase was the intervention itself. The workshops for intervention group coaches took place three times in the span of 9 weeks, and for 12 weeks, coaches practiced exercises during training with the participants/child athletes. Prior to the second and third workshop and after the last workshop, the coaches gave feedback on practicing the i7W model (template mentioned above). The control group of coaches received no workshop at that time. The final phase took place in the laboratory, where the children underwent the same tests as prior to the intervention. Coaches were then sent anonymized reports based on a preliminary description (descriptive statistics) of psychological characteristic and psychomotor performance of each child.

### 2.3. Psychological Tools

Selected methods were used by the authors to estimate psychological characteristics and psychomotor performance of participants/child athletes. All the questionnaires were appropriate for children and completed by children in the presence of a psychologist. 

### 2.4. Psychomotor Performance-Vienna Test System

The authors used two tests: The Determination Test for Kids (DTKI) to measure reactive stress tolerance, reaction speed, and attention deficits in situations requiring responses to rapidly changing stimuli, as well as the reaction time test (RT), which measures multifaceted reaction time. 

### 2.5. Sport Competition Anxiety Test (SCAT) and Competitive State Anxiety Inventory-2 (CSAI-2RD)

These tests were used to measure trait and state sports anxiety and its intensity [43,44]. Both methods were adapted for Polish conditions by Borek-Chudek [45,46]. The SCAT is a 15-item questionnaire containing statements about somatic anxiety (e.g., My body feels tense), worry (e.g., I worry that I will not play my best), and concentration disruption (e.g., I lose focus on the game). The items are rated on a three-point scale. The CSAI-2RD consists of a 14-item scale containing cognitive anxiety statements (e.g., I was confident because, in my mind, I pictured myself reaching my goal) and somatic anxiety statements (e.g., My heart was racing). The competitive state anxiety is rated on a four-point scale, and anxiety intensity is estimated on the five-point scale. Both tests have been shown to have acceptable psychometric properties [45,46]. In the current study, the coefficient alphas for the footballers and gymnasts (respectively) were as followed: trait anxiety 0.74, 0.64; state anxiety 0.76, 0.89; state anxiety self-reflection 0.87, 0.88. 

### 2.6. Sport Motivation Scale-6 (SMS-6)

A Polish adaptation of the SMS-6 by Blecharz et al. (2015) was used to measure intrinsic and extrinsic motives for sport activity [47]. There are six kinds of motivation measured: intrinsic motivation (e.g., For the excitement I feel when I am really involved in the activity), amotivation (e.g., I do not seem to be enjoying my sport as much as I previously did), and four types of extrinsic motivation (e.g., Because it is one of the best ways to maintain good relationships with my friends; To show others how good I am at my sport). For the purpose of the study, the authors modified the test and shortened the performance time of the test, taking into consideration the fact that children’s concentration of attention is weaker than that of adults. The young participants did not assess the motives with a seven-point Likert scale but instead chose the motives they preferred. We opted for this modification when we found, during familiarization to the study, that the children selected either the first or seventh point of each scale. Each of the five motivations in each of the six subscales was scored as 0 or 1, and the score for each subscale was the mean of its five values. The scale has been shown to be valid and reliable in a sample of Polish athletes [47]. In the current study, the Cronbach’s alpha for the footballers and gymnasts (respectively) for mean motivation was 0.73 and 0.84. 

### 2.7. The Polish Coach Athlete–Relationship Questionnaire—Version for Athlete (PICART-Q Athlete)

A Polish adaptation of the coach–athlete relationship questionnaire was used. The questionnaire has 11 items and consists of three subscales: closeness (e.g., I trust my coach), commitment (e.g., My coach is close to me), and complementarity (e.g., When I practice with my coach, I am ready to do my best) estimated on the seven-point Likert scale. The scales, in the previous studies, have shown acceptable Cronbach’s alpha [48]. The alphas for the footballers and gymnasts for the mean coach–athlete relationship were 0.91 and 0.95, respectively, in our study.

### 2.8. Ethics

Written informed consent from coaches and from parents or guardians of the children was a prerequisite to participate in the study. The project was approved by the Bioethics Committee at the District Medical Chamber in Gdansk, Poland (decision for the project no. KB-13/17).

### 2.9. Statistics

Statistical analyses were performed with the IBM SPSS Statistics, Version 24.0 (IBM, Armonk, New York, NY, USA). Repeated measures ANOVAs were used for variables with a normal distribution, and nonparametric test was used for variables with non-normal distribution. Association between variables were tested with Pearson’s and Spearman’s correlations. Multivariable analyses were performed using a linear regression model.

## 3. Results

Firstly, we aimed to evaluate the differences in scores obtained in Vienna Test System (VTS): the Determination Test for Kids (DTKI) and the reaction time test (RT), as well as Sport Motivation Scale-6 (SMS-6) between groups in the two assessments. Yet, before that, we performed the Kolmogorov–Smirnov test (K–S test) in order to verify the normality of the distributions of the variables in our research. The K–S test revealed that the distributions of some variables differ significantly from normal distribution and thus we decided to use nonparametric testing (with Mann–Whitney U-test and Wilcoxon T-test) in these cases (variables from the first and from the second assessment: VTS DTKI mistakes, VTS DTKI omitted stimuli, VTS RT correct responses, VTS RT lack of reactions, VTS RT mistakes as well as all SMS-6 variables). For all other assessments of the differences, we used parametric testing with ANOVA. Groups did not differ significantly in any of the VTS variables, but they did in some aspects of sport motivation in both the first assessment (baseline) and the second measurement (post-test). We also found several significant between-measurements differences (see Table 2 for the details).

Then, we obtained scores representing the change between the measurements for each participant in VTS and SMS-6 assessments and evaluated the differences between groups with ANOVA (the procedure was preceded by the K–S testing in order to select the correct statistics to evaluate the differences). The significant difference between intervention and control group was found for the change in the VTS DTKI correct responses as well as SMS-6 change in intrinsic motivation and change in external regulation. The details are presented in Table 3.

Next, we evaluated the relation of the changes in VTS and SMS-6 assessments with the initial coach–athlete relationship measures (PICART: closeness, commitment, and complementarity) as well as sport state and trait anxiety (SCAT and CSAI-2RD) with the Pearson’s and Spearman’s correlation coefficients. Of note, there were no significant differences in PICART SCAT and CSAI-2RD between groups and between the two assessments (see Table 4 for the details).

In the intervention group, we found significant positive correlation between the change in the VTS RT mean reaction time and PICART closeness (r = 0.45, *p* = 0.016), negative between trait anxiety and change in the VTS RT mean reaction time (r = −0.44, *p* = 0.023), and positive correlation between change in the SMS-6 introjection regulation and PICART complementarity (r = 0.41, *p* = 0.041). In the control group we found positive correlation between PICART commitment and change in the SMS-6 identified regulation (r = 0.42, *p* = 0.043), as well as negative correlation between the sport trait anxiety and the change in SMS-6 internal motivation (r = −0.55, *p* = 0.003) and the change in SMS-6 integrated regulation (r = −0.39, *p* = 0.043).

In the final part of the analyses, on the basis of the previously obtained results, we evaluated the predictors of the VTS and SMS-6 post-test scores as dependent variables using the entrance or stepwise regression analyses. A dichotomic intervention group variable, pretest measures of PICART dimensions, and pretest SCAT anxiety factors were entered in the first step of the regression model, and the pretest PICART and SCAT measures per intervention-group interaction effects were tested in the second step of the regression model with the stepwise method. During the evaluation of the predictors of the change in VTS RT mean reaction time, at the second regression step, the PICART closeness per group interaction was a significant predictor, excluding other interaction terms from further analysis, explaining 40.8% of the variance in the VTS RT post-test mean reaction time (R2 = 0.408, *F* = 3.443, *p* = 0.004, b = 18.729). In the second analysis, with the post-test SMS-6 introjected regulation score as explained variable, we obtained insignificant results (*p* > 0.05). However, we found that intervention condition per PICART commitment interaction effect does predict the post-test SMS-6 identified regulation scores (R2 = 0.349, *F* = 2.682, *p* = 0.018, b = −0.079). We found that intervention condition per SCAT-sport trait anxiety interaction effect predicts the change in variance explained for the SMS-6 integrated motivation post-test (R2 = 0.256, ΔF = 4.856, *p* = 0.033, b = 0.040); as well as that SCAT- sport trait anxiety interaction effect does predict the change in variance explained for the SMS-6 internal motivation post-test (R2 = 0.209, ΔF = 5.939, *p* = 0.019, b = 0.076).

## 4. Discussion

The authors of the study had two goals. Firstly, to investigate the impact of workshops for coaches on their young athletes’ autonomous motivation, psychomotor performance, and positive affect while performing in a sports context, and the changes in an Athlete-Coach positive relationship. Secondly, to determine if such aspects as competitive anxiety and selected characteristics of Athlete-Coach relationship will predict the change in motivation and psychomotor performance. The results showed significant differences between the intervention and control group in the VTS DTKI correct responses, as well as SMS-6 intrinsic motivation and external regulation. After implementing the i7W model, the correct reactions in psychomotor performance increased in children whose coaches implemented the model. The i7W drills and tasks, which coaches were asked to perform on children during the experiment, were created to increase the self-determination aspects, such as competence, as well as relatedness and autonomy. We assume that psychomotor performance as the component of the overall sports performance of children could be changed because of the character of i7W model drills and tasks, of which one of the aims is to increase self-determination. In a 3-year longitudinal study on French national tennis players aged 13 and 14 years, Gillet et al. (2009) proved that self-determined motivation positively influenced performance [12]. On the other hand, the internal motivation and external regulation did not increase, but their levels decreased by significantly less than those among children with no access to the i7W model. That result, especially the change of intrinsic motivation, seems to be interesting and, at the same time, worrisome. We could assume from the results that participants have been experiencing less pleasure from sports participation. Children from the experimental group received more positive, autonomy-supportive coaching through the exercises from the i7W model (inspire, explain, expect, support, reward athlete to grow, and win). That could stop the intrinsic motivation from falling as much as among control group children whose coaches could express a different style of coaching, maybe a more controlling one. Standage and Ryan (2020) underline that the controlling coaching style could promote ill-being by using negative feedback, control, and pressure strategies [49]. In the studies on young swimmers, Alvarez et al. (2021) showed that a controlling coaching style causes boredom and, through that, influences the burnout symptoms [50]. In addition, in their paper, Witt and Dangi (2018) underlined that one of the children and youths’ leading reasons to drop out of sport is not having fun, being bored, and having negative feelings toward their coach and team, as well as experiencing parental pressure [51]. Wilczyńska et al. (2021) showed in the pilot study that i7W model tools and drills could influence some factors of wellbeing in children practicing sport [38]. Therefore, we think that coaches’ implementation of the i7W model in training and competition could have a protective effect on the intrinsic motivation of their athletes. Moreover, in our opinion, sports specialization among studied children could lower their intrinsic motivation. Brenner et al. (2019), in their research review on the implications of sport specialization in pediatric athletes, wrote that early sport specialization could harmfully affect the mental health and functioning of young athletes and increase the risk for burnout [52].

The analysis of the predictors revealed that the i7W model intervention used in this study might be effective for enhancing psychomotor performance (VTS) and motivation (SMS-6) while considering components of Athlete-Coach relation (PICART-Q) and the level of sport anxiety (SCAT) as moderators. The PICART-Q closeness moderated the change in simple reaction time in the VTS RT test, and PICART commitment moderated post-test SMS-6 identified regulation level. In addition, SCAT (sports trait anxiety) predicted the post-test SMS-6 integrated motivation and internal motivation changes in the experimental group. We assume that those athletes who are less close and less committed to the coach with a higher level of sports trait anxiety could benefit more from the i7W drills and recommendations, especially considering psychomotor functioning and autonomous motivation. For example, the quality of the emotional interdependence is developed by the tasks of i7W support, mutual respect through i7W to expect maximization of sport development and results through long-term relationship, i7W grow and safe, comfortable psychological attachment among athlete and coach, i7W support drills. As Vella et al. (2013) underline, the quality of the coach–athlete relationship influences different spheres of psychological functioning of young athletes [53]. We know from the studies that an emotional bond with the athlete can tremendously impact young athletes’ training and sports performance. Coaches who acknowledge athletes’ feelings and perspectives develop positive and secure attachment, the solid foundation of long-term sport persistence [53,54,55].

As for the relations, the same importance should be placed on the emotions of the youth athletes. For instance, the high level of sports trait anxiety in athletes causes them to underestimate their abilities to overcome situational demands, perceive competition as more threatening, and, thus, make them more susceptible to amotivation and burnout [56,57,58]. As we underlined previously in the text, children who undergo negative emotions will not develop appropriately in the sports context. The lack of sport enjoyment is a threat to intrinsic motivation, the leading component of persistence in sport. Therefore, educational interventions and workshops for coaches with the tools and drills of the i7W model could significantly create mentally and physically healthy athletes with long-term careers. The i7W model for coaches focuses on a holistic approach to an athlete’s career, especially on a positive approach to children and youth sports. The model’s authors underline that a coach who works with young athletes should plan a training session with thoughtful consideration of children’s and youth’s capacities. The coach who carefully ensures a positive coaching environment and conditions is the one who makes children thrive in sports and outside while maintaining joy and satisfaction [37].

Despite its valuable results, there are limitations of this study. First of all, as Vella (2021) underlines, the way to improve the quality of the interventions for the mental health of youth athletes is to intervene at multiple levels [59]. More effective are those interventions which consider working with athletes, coaches, and parents. In this particular study, we focused our research on children only. Future studies should consider i7W workshops for both coaches and parents, as well as children separately. Moreover, the change in coach and parent behavior should be investigated due to the i7W model implementation. Such aspects as personality, coaching style, and, especially, prior experience with the i7W model are also reasonable to consider since some coaches in Poland could be familiar with the principles. In addition, more careful observation of the coaches during training sessions and competitions and one-on-one meetings with coaches could reveal how well they implement the principles of the i7W model.

## 5. Conclusions

Coach workshops based on the i7W model have been previously used among adult athletes to enhance the sports experience. The authors of the current study decided to investigate workshops with coaches practicing with child athletes, which is undoubtedly a novelty. The study revealed several significant results. Firstly, there were substantial differences between intervention and control group children in correct responses of psychomotor test and the level of intrinsic motivation after implementing i7W workshops for coaches. The experimental group was characterized with more correct responses and a lower decrease of intrinsic motivation than the control group. Secondly, the closeness of the Athlete-Coach relationship moderated the change in simple reaction time, and the commitment aspect of relation moderated the change in identified motivation. In addition, sport competitive trait anxiety was a predictor of integrated and intrinsic motivation change. Even though the study showed some significant effects of the i7W model on selected psychological factors of children practicing sport and the predictors of those changes, the model still requires future research. A more significant sample of young athletes is required to investigate the impact of the i7W model.

## Figures and Tables

**Table 1 ijerph-19-03462-t001:** Workshop topics (selected dimensions from i7W concept) and examples of the tools and activities introduced during the workshop.

Workshop; Topics	Activities
First workshop; inspire, explain	“Pump them up”—Prepare positive coach speech before your athletes’ competition. “Strengthening analysis”—Conduct a post-match analysis with the athletes, focusing on the positive aspects of the game.
Second workshop; expect, support	“Show your hand”—At the end of each week analyze simple goals that youth athletes formulated at the beginning of the week. Stimulate and help them to find answers to the following questions: How did I reach my goal? If not, what should I do differently in the future? “Positive bakes”—During training divide children into pairs. Each child from the pair is tasked throughout the training to provide verbal and non-verbal support to teammate, e.g., giving high-five after a good performance, or a good word when teammate made a mistake. At the end of the training, ask the pair whether they felt the support of the teammate.
Third workshop; reward, appreciate	“Good mistake”—Praise athletes despite them making mistakes. Show other youth athletes that creativity and courage is more important than making a single mistake. Highlight the fact that the biggest mistake one can make is not looking for the solution. “Master T-shirt”—Introduce a special T-shirt for player who has shown the greatest commitment during training sessions.

Each workshop described a few dimensions that contribute to “I grow and I win” of youth athletes. During each workshop, the coaches were taught practical applications and were provided with activities and a timetable to implement the i7W model. The coaches were also provided with a template to record the activities they implemented. The workshops were organized in 6 h meetings over a 9-week period (one workshop every 3 weeks). To control the implementation of the activities, the coaches sent the template to the coordinator of the study the day before the second, and the third workshop, and 3 weeks after the last workshop.

**Table 2 ijerph-19-03462-t002:** The differences in the mean scores obtained for the VTS and SMS-6 variables between groups (intervention vs. control) and between measurements (baseline vs. post-test).

	Baseline			Post-Test			
Variable	Intervention (Mean ± SD)	Control (Mean ± SD)	Between-Group Difference	Intervention (Mean ± SD)	Control (Mean ± SD)	Between-Group Difference	Between-Measurements Difference
**VTS:**							
**DTKI correct responses**	193.11 ± 25.50	199.52 ± 22.97	*F* = 0.996; *p* = 0.323	223.79 ± 32.11	215.24 ± 27.80	*F* = 1.156; *p* = 0.287	**I ^a^: *F* = 7.830; *p* = 0.009** **C ^b^: *F* = 64.696; *p* < 0.001**
**DTKI mistakes**	24.57 ± 24.68	21.72 ± 16.79	*Z* = −0.208; *p* = 0.836	28.36 ± 30.77	28.86 ± 23.50	*Z* = −0.280; *p* = 0.780	I: *Z* = 0.866; *p* = 0.387 C: *Z* = 1.899; *p* = 0.058
**DTKI omitted stimuli**	17.32 ± 10.43	19.90 ± 13.63	*Z* = −0.632; *p* = 0.528	16.61 ± 11.43	23.59 ± 14.18	*Z* = −1.901; *p* = 0.057	I: *Z* = −0.648 *p* = 0.517 C: *Z* = 1.527; *p* = 0.127
**RT mean reaction time**	479.21 ± 66.54	501.41 ± 82.99	*F* = 1.236; *p* = 0.271	452.43 ± 68.83	457.34 ± 68.25	*F* = 0.073; *p* = 0.788	**I: *F* = 6.610; *p* = 0.016** **C: *F* = 7.644; *p* = 0.010**
**RT correct responses**	15.64 ± 0.83	15.83 ± 0.47	*Z* = −0.796; *p* = 0.426	15.93 ± 0.26	15.90 ± 0.31	*Z* = 0.423; *p* = 0.672	I: *Z* = 1.613 *p* = 0.107 C: *Z* = 0.707; *p* = 0.480
**RT lack of reactions**	0.32 ± 0.82	0.17 ± 0.47	*Z* = 0.478; *p* = 0.632	0.07 ± 0.26	0.10 ± 0.31	*Z* = −0.423; *p* = 0.672	I: *W* = 1.406; *p* = 0.160 C: *W* = −0.707; *p* = 0.480
**RT incomplete reactions**	0	0	NA	0	0	NA	NA
**RT mistakes**	0.46 ± 2.08	0.03 ± 0.19	*Z* = 1.082; *p* = 0.279	0	0	NA	NA
**SMS-6:**							
**Intrinsic motivation**	0.72 ± 0.28	0.90 ± 0.16	***Z* = −2.510;** ***p* = 0.012**	0.69 ± 0.63	0.54 ± 0.35	*Z* = 0.680; *p* = 0.497	I: *Z* = −1.103; *p* = 0.270 **C: *Z* = −3.775;** ***p* < 0.001**
**Integrated regulation**	0.87 ± 0.19	0.88 ± 0.21	*Z* = −0.584; *p* = 0.559	0.77 ± 0.30	0.60 ± 0.33	***Z* = 2.001;** ***p* = 0.045**	I: *Z* = −1.762; *p* = 0.078 **C: *Z* = −3.563; *p* < 0.001**
**Identified regulation**	0.73 ± 0.25	0.76 ± 0.21	*Z* = −0.0275; *p* = 0.783	0.63 ± 0.26	0.51 ± 0.30	*Z* = 1.226; *p* = 0.220	I: *Z* = −1.740; *p* = 0.082 **C: *Z* = −3.517; *p* < 0.001**
**Introjected regulation**	0.74 ± 0.24	0.67 ± 0.27	*Z* = 0.918; *p* = 0.359	0.60 ± 0.33	0.56 ± 0.65	*Z* = 1.442; *p* = 0.149	I: *Z* = −1.912; *p* = 0.056 **C: *Z* = −2.331; *p* = 0.020**
**External regulation**	0.42 ± 0.33	0.64 ± 0.32	***Z* = −2.382;** ***p* = 0.017**	0.38 ± 0.32	0.26 ± 0.29	*Z* = 1.357; *p* = 0.175	I: *Z* = −0.583; *p* = 0.560 **C: *Z* = −3.690; *p* < 0.001**
**Amotivation**	0.07 ± 0.16	0.08 ± 0.18	*Z* = 0.011; *p* = 0.991	0.05 ± 0.12	0.08 ± 0.19	*Z* = −0.072; *p* = 0.947	I: *Z* = −0.351; *p* = 0.726 C: *Z* = 0.071; *p* = 0.943

Note. In case of variables with the distribution close to normal distribution, we used parametric testing with ANOVA, and in case of variables with a distribution significantly different from the normal distribution we used nonparametric testing with Mann–Whitney U-test (between-group differences) and Wilcoxon T-test (between-measurements difference). I ^a^—Intervention group, n = 28. C ^b^—Control group, n = 29.

**Table 3 ijerph-19-03462-t003:** The differences in the change of VTS scores in time between intervention and control group.

Variable	Intervention (Mean ± SD)	Control (Mean ± SD)	Between-Group Difference
**VTS:**			
**Change in DTKI correct responses**	30.69 ± 20.18	15.72 ± 30.26	***F* = 4.782;** ***p* = 0.033**
**Change in DTKI mistakes**	3.79 ± 23.81	7.14 ± 18.20	*F* = 0.358; *p* = 0.552
**Change in DTKI omitted stimuli**	−0.71 ± 7.62	2.69 ± 11.85	*F* = 2.761; *p* = 0.102
**Change in RT mean reaction time**	−26.79 ± 55.13	−44.07 ± 85.83	*F* = 0.812 *p* = 0.372
**Change in RT correct responses**	0.29 ± 0.90	−0.07 ± 0.53	*Z* = 0.913; *p* = 0.361
**Change in RT lack of reactions**	−0.25 ± 0.89	0	NA
**Change in RT incomplete reactions**	0	0	NA
**Change in RT mistakes**	−0.46 ± 2.08	−0.03 ± 0.19	*Z* = −1.082; *p* = 0.279
**SMS-6:**			
**Change in Intrinsic motivation**	−0.04 ± 0.70	−0.36 ± 0.38	***Z* = 1.992;** ***p* = 0.046**
**Change in Integrated regulation**	−0.10 ± 0.28	−0.28 ± 0.33	*Z* = 1.943; *p* = 0.052
**Change in Identified regulation**	−0.11 ± 0.28	−0.25 ± 0.28	*Z* = 1.328; *p* = 0.184
**Change in Introjection regulation**	−0.14 ± 0.36	−0.11 ± 0.64	***Z* = 0.637;** ***p* = 0.524**
**Change in External regulation**	−0.04 ± 0.43	−0.38 ± 0.38	*Z* = 3.016; *p* = 0.003
**Change in Amotivation**	−0.02 ± 0.20	−0.01 ± 0.25	*Z* = −0.278; *p* = 0.781

Note. Intervention group: n = 28; control group: n = 29. In case of variables with distribution close to normal distribution we used parametric testing with ANOVA, and in case of variables with a distribution significantly different from normal distribution we used nonparametric testing with Mann–Whitney U-test.

**Table 4 ijerph-19-03462-t004:** The differences in the mean scores obtained for the PICART, SCAT and CSAI-2RD variables between groups (intervention vs. control) and between measurements (baseline vs. post-test).

	Baseline			Post-Test			
Variable	Intervention (Mean ± SD)	Control (Mean ± SD)	Between-Group Difference	Intervention (Mean ± SD)	Control (Mean ± SD)	Between-Group Difference	Between-Measurements Difference
**PICART:**							
**PICART closeness**	24.32 ± 3.28	26.21 ± 1.64	*Z* = −1.954; *p* = 0.051	22.96 ± 7.16	23.52 ± 6.21	*Z* = −0.389; *p* = 0.698	I ^a^: *Z* = −0.273; *p* = 0.785 C ^b^: *Z* = −1.742; *p* = 0.082
**PICART commitment**	15.89 ± 2.70	16.96 ± 2.26	*F* = 2.339; *p* = 0.132	14.46 ± 5.39	14.90 ± 4.90	*Z* = −0.241; *p* = 0.810	I: *Z* = −0.957 *p* = 0.338 **C: *Z* = −2.322; *p* = 0.020**
**PICART complementarity**	22.75 ± 3.57	25.58 ± 2.13	***Z* = −2.908;** ***p* = 0.004**	21.32 ± 7.20	22.48 ± 6.16	*Z* = −0.634; *p* = 0.526	I: *Z* = −0.703 *p* = 0.482 **C: *Z* = −2.359; *p* = 0.018**
**ANXIETY:**							
**SCAT (trait anxiety)**	19.93 ± 5.60	17.67 ± 4.88	*Z* = 1.528; *p* = 0.127	18.11 ± 6.37	17.90 ± 6.74	*Z* = 0.408; *p* = 0.683	I: *Z* = −1.425; *p* = 0.154 C: *Z* = −0.244; *p* = 0.807
**CSAI-2RD (state anxiety)**	29.93 ± 8.48	26.22 ± 5.80	*F* = 3.511; *p* = 0.067	28.04 ± 7.63	26.28 ± 8.08	*F* = 0.802; *p* = 0.375	I: *F* = 1.654; *p* = 0.210 C: *F* = 0.003; *p* = 0.956

Note. In case of variables with the distribution close to normal distribution we used parametric testing with ANOVA, and in case of variables with a distribution significantly different from the normal distribution we used nonparametric testing with Mann–Whitney U-test (between-group differences) and Wilcoxon T-test (between-measurements difference). I ^a^—Intervention group, n = 28. C ^b^—Control group, n = 29.

## Data Availability

We have not made the data available.

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
