# Peer review of "Dimensions of Athlete-Coach Relationship and Sport Anxiety as Predictors of the Changes in Psychomotor and Motivational Welfare of Child Athletes after the Implementation of the Psychological Workshops for Coaches"

_ijerph, 2022, doi:10.3390/ijerph19063462_

Round 1

Reviewer 1 Report

The idea of the research is good and I would like to congratulate the authors for it, however, I would like to detail some things to consider.

The introduction is too long

Materials and method:

There are numbers written with numbers and others with letters. i recommend following the same pattern.

The sample is not very homogeneous since soccer athletes work in teams and gymnasts work more individually, so motivation and stress may be different and difficult to compare.

Results: include everything related to statistics at the end of the method.

I recommend changing end of study to post-test.

I would like to know if any variable was measured that was not so subjective (some biomarker). 

Author Response

Dear reviewer, we greatly appreciate your work. Thank you for your effort and valuable comments, which help us grow in our work and develop our abilities.

Point raised by referee

Response by author

Location in text:

Page and paragraph reference

Specific comments

The introduction is too long

We modified and shortened the Introduction

Materials and method:

There are numbers written with numbers and others with letters. i recommend following the same pattern.

The sample is not very homogeneous since soccer athletes work in teams and gymnasts work more individually, so motivation and stress may be different and difficult to compare.

Our understanding is that numbers less than 10 are spelt out, unless they are shown with units (nine people, 9 kg). "Children nine-twelve years old" just doesn't look right”. So we decided not to change this aspect.

We realize the difference; however, we also have team competitions in such individual sports as gymnastics. During the year’s performance cycle in Poland, there have been gymnastic national team championships in different age categories where young child athletes represent, compete and score as a team. We have the same kind of competitions in other individual disciplines, such as tennis, wherein the senior category, we have famous Davis Cup or Fed Cup competitions. On the other hand, children practice their sports disciplines in groups, even in individual ones. As Bruner and Benson (2018) indicate in their study, the social identity to the group is vital in both team and individual sports and influences individual outcomes such as self-worth, commitment, personal and social skills. Rhythmic gymnasts also perform in groups doing same gymnastic configurations.

Bruner, M. W., & Benson, A. J. (2018). Evaluating the psychometric properties of the Social Identity Questionnaire for Sport (SIQS). Psychology of Sport and Exercise, 35, 181–188. https://doi.org/10.1016/j.psychsport.2017.12.006 

Also, we focused on team relations as athlete-coach relationships and exercises we used from the i7W model have been created to develop positive athlete-coach relationships. We underlined that in the Introduction chapter : “The results showed that athletes whose coaches practiced the i7W model in eight work-shops (lasting one and half hours each) estimated their relationships with coaches to be significantly stronger and their self-confidence and group cohesion to be higher.”

Moreover, the athlete and coach team relation is an important aspect, especially in youth sport. It does not matter if child athletes practice team or individual sports, the relation with a coach plays a significant role in developing young athletes. That variable was one of the most essential of our study.

However, to control for differences due to gender or discipline, it was statistically tested that there were no differences in the composition of the experimental and control groups.

“A Pearson's Chi-square test was performed to check for group differences by gender or discipline, finding that the experimental and control groups were balanced by gender or discipline (χ2 = 2.12, p = .14).”

Lines: 147-150

Lines:179-181

Results: include everything related to statistics at the end of the method.

I recommend changing end of study to post-test.

The change is done

I would like to know if any variable was measured that was not so subjective (some biomarker).

No biomarkers were measured however we estimated balance (motor ability) differences among participants which was the topic of other article from this project, which we recommend (Wilczyńska, et al., 2021)

Wilczyńska, D., Łysak-Radomska, A., Podczarska-Głowacka, M., Krasowska, K., Liedtke, E., Walentukiewicz, A., Lipowski, M., & Skrobot, W. (2021a). Effect of workshops for coaches on the motor ability of balance in children practicing sports in late childhood. BMC Sports Science, Medicine and Rehabilitation, 13. https://doi.org/10.1186/s13102-021-00388-9 

Reviewer 2 Report

Review of POL_1
The reviewed work is undoubtedly interesting. At the same time, a number of important questions arise. First of all, what is the novelty of the work. In the conclusions - of which the authors have two, neither the first conclusion nor the second conclusions are new. The fact that after the i7W seminars, differences between the experimental group and the control group were revealed is at the very heart of the proposed seminars and is not new. Perhaps the novelty is related to the age of the athletes involved in the survey – but the authors do not indicate this clearly enough in the text. The second conclusion of the survey is a particular description of a certain positive effect of the training procedure used, and is a rather particular phenomenological feature that the authors described as a whole on a relatively small sample. 
There are also questions about the sample itself, since the authors have sufficiently described their approach to the selection of sports, and, accordingly, the groups of young athletes they surveyed. The authors rightly point out that they chose gymnastics and football because of their early specialization and the exclusion of other sports, which means that participants could not engage in other sports disciplines. At the same time, with this approach, two game sports are combined into one group – however, football is a team game, and gymnastics is an individual sport seems very doubtful. It is known that football is a team, situational, game sport. At the same time, gymnastics is a highly coordinated sport. In addition, it is not clear what was meant, i.e. what exactly did the children do - projectile gymnastics, floor exercises, support jump, or was it rhythmic gymnastics at all?

There is no information in the article about the amount of training loads during the period of the study, they could give the amount of training in hours per week / month, including for children engaged in different sports. The authors apparently have no understanding in which period of the annual training cycle, preparatory / competitive / recovery, the study was conducted?
In the results of the research it is not clear who was in which groups, according to gender characteristics and sports, everything is mixed into one pile.
In addition, the text does not indicate how many representatives of different sexes were in sports - without understanding this fact, all other conclusions of the authors become generally unfounded. In the materials and methods, the authors do not represent groups in gender differences, but in the abstract it is shown that the groups were of boys and girls. It is known that the indicators evaluated in the peer-reviewed work may have gender differences, since the motivational aspect in boys and girls is usually formed under the influence of different factors, the anxiety manifested is also due to "problems" in different spheres. Girls, as a rule, are more emotional and prone to internal and social experiences - appearance, position in the group, attractiveness, while boys are more often focused on external factors - success, recognition, affirmation of their own position. Stress factors have different manifestations in groups of boys and girls. When conducting testing, these differences should be taken into account without fail in order to obtain reliable results
As an additional editorial criticism, we can note a very large introduction to the article, in which the approaches to conducting the study are described in too much detail, but the alleged novelty of the work is completely insufficiently substantiated. The management is too stretched, psychological concepts are described in great detail and in detail, I think it needs to be shortened, reflecting what is described in the results.

Author Response

Dear reviewer, we greatly appreciate your work. Thank you for your effort and valuable comments, which help us grow in our work and develop our abilities.

References

Bruner, M. W., & Benson, A. J. (2018). Evaluating the psychometric properties of the Social Identity Questionnaire for Sport (SIQS). Psychology of Sport and Exercise, 35, 181–188. https://doi.org/10.1016/j.psychsport.2017.12.006

Hyde, J. S. (2005). The gender similarities hypothesis. American Psychologist, 60(6), 581–592. https://doi.org/10.1037/0003-066X.60.6.581

Hyde, J. S. (2014). Gender similarities and differences. Annual Review of Psychology, 65(1), 373–398. https://doi.org/10.1146/annurev-psych-010213-115057

Visek, A. J., Mannix, H., Chandran, A., Cleary, S. D., McDonnell, K. A., & DiPietro, L. (2020). Toward understanding youth athletes’ fun priorities: An investigation of sex, age, and levels of play. Women in Sport & Physical Activity Journal, 28(1), 34–49.

Wilczyńska, D., Łysak-Radomska, A., Podczarska-Głowacka, M., Krasowska, K., Liedtke, E., Walentukiewicz, A., Lipowski, M., & Skrobot, W. (2021a). Effect of workshops for coaches on the motor ability of balance in children practicing sports in late childhood. BMC Sports Science, Medicine and Rehabilitation, 13. https://doi.org/10.1186/s13102-021-00388-9

Wilczyńska, D., Łysak-Radomska, A., Podczarska-Głowacka, M., Skrobot, W., Krasowska, K., Liedtke, E., Dancewicz, T., Lipińska, P., & Hopkins, W. (2021b). The effectiveness of psychological workshops for coaches on well-being and psychomotor performance of children practicing football and gymnastics. Journal of Sports Science and Medicine, 20, 586–593. https://doi.org/10.52082/jssm.2021.586

Zell, E., Krizan, Z., & Teeter, S. R. (2015). Evaluating gender similarities and differences using metasynthesis. American Psychologist, 70(1), 10–20. https://doi.org/10.1037/a0038208

Point raised by referee

Response by author

Location in text:

Page and paragraph reference

Specific comments

The reviewed work is undoubtedly interesting. At the same time, a number of important questions arise. First of all, what is the novelty of the work. In the conclusions - of which the authors have two, neither the first conclusion nor the second conclusions are new. The fact that after the i7W seminars, differences between the experimental group and the control group were revealed is at the very heart of the proposed seminars and is not new. Perhaps the novelty is related to the age of the athletes involved in the survey – but the authors do not indicate this clearly enough in the text. The second conclusion of the survey is a particular description of a certain positive effect of the training procedure used, and is a rather particular phenomenological feature that the authors described as a whole on a relatively small sample.

We have added two sentence in the Conclusion to add the information on the novelty of the current study and the need for bigger sample: “Coach workshops based on the i7W model have been previously used among adult athletes to enhance the sports experience. The authors of the current study decided to investigate workshops with coaches practicing with child athletes, which is undoubtedly a novelty” and “A more significant sample of young athletes is required to investigate the impact of the i7W model.”

Lines 468-471

Lines 481-482

There are also questions about the sample itself, since the authors have sufficiently described their approach to the selection of sports, and, accordingly, the groups of young athletes they surveyed. The authors rightly point out that they chose gymnastics and football because of their early specialization and the exclusion of other sports, which means that participants could not engage in other sports disciplines. At the same time, with this approach, two game sports are combined into one group – however, football is a team game, and gymnastics is an individual sport seems very doubtful. It is known that football is a team, situational, game sport. At the same time, gymnastics is a highly coordinated sport. In addition, it is not clear what was meant, i.e. what exactly did the children do - projectile gymnastics, floor exercises, support jump, or was it rhythmic gymnastics at all?

We realize the difference; however, we also have team competitions in such individual sports as artistic gymnastics. During the year’s performance cycle in Poland, there have been gymnastic national team championships in different age categories where young athletes represent, compete and score as a team. We have the same kind of competitions in other individual disciplines, such as tennis, wherein the senior category, we have famous Davis Cup or Fed Cup competitions. Rhythmic gymnasts also perform in groups doing same gymnastic configurations. As Bruner and Benson (2018) indicate in their study, the social identity to the group is vital in both team and individual sports and influences individual outcomes such as self-worth, commitment, personal and social skills. It does not matter if child athletes practice team or individual sports, the relation with a coach plays a significant role in developing young athletes. That variable was one of the most essential of our study.

On the other hand, the sample is homogenous in the way it has almost the same number of individual and team sports representatives in both experimental and control groups. The number of boys and girls was also the same.

Gymnasts were from rhythmic and artistic gymnastics, what we added in the text

Line 162

There is no information in the article about the amount of training loads during the period of the study, they could give the amount of training in hours per week / month, including for children engaged in different sports. The authors apparently have no understanding in which period of the annual training cycle, preparatory / competitive / recovery, the study was conducted?

We added required information in the Material and Methods (subchapter Participants): “(...)and week training loads (4,9 ± 1,4 and 5,1 ± 1.6 training per week; 121 ± 43,5 and 131,7 ± 45,7minutes per training). Children were investigated in Autumn and early Wintertime, during the school semester, preparatory and competitive period for children practicing football and gymnastics in Poland”.

Lines:178-179

Lines: 181-183

In the results of the research it is not clear who was in which groups, according to gender characteristics and sports, everything is mixed into one pile.

In addition, the text does not indicate how many representatives of different sexes were in sports - without understanding this fact, all other conclusions of the authors become generally unfounded. In the materials and methods, the authors do not represent groups in gender differences, but in the abstract it is shown that the groups were of boys and girls. It is known that the indicators evaluated in the peer-reviewed work may have gender differences, since the motivational aspect in boys and girls is usually formed under the influence of different factors, the anxiety manifested is also due to "problems" in different spheres. Girls, as a rule, are more emotional and prone to internal and social experiences - appearance, position in the group, attractiveness, while boys are more often focused on external factors - success, recognition, affirmation of their own position. Stress factors have different manifestations in groups of boys and girls. When conducting testing, these differences should be taken into account without fail in order to obtain reliable results

The gender differences was not the variable considered in the study. We did that on purpose as long as children were in the late childhood stage of development, and all of them were before puberty. Significant gender differences starts during and after puberty. Moreover, the results of our other analysis on the same group, while considering gender differences in psychological well-being characteristics showed similar effects of the workshops for boys and girls (Wilczyńska, et al., 2021b). We also decided on the same procedure in other published study (Wilczyńska, et al., 2021a) on the same group of child athletes, where we measured differences in balance motor ability after the implementation of the i7W model for coaches. Both control and intervention groups were mixed with boys and girls, not considering gender variable. Also there has been a gender similarities hypothesis when studying females and males across psychosocial domains. According to the research, females and males (boys and girls) are consistently found to be more alike than different; and, in the case of differences, the magnitude of those differences is quite small, especially in sport domain (Hyde, 2005, 2014; Visek et al., 2020; Zell et al., 2015).

However, to control for differences due to gender or discipline, it was statistically tested that there were no differences in the composition of the experimental and control groups.

“A Pearson's Chi-square test was performed to check for group differences by gender or discipline, finding that the experimental and control groups were balanced by gender or discipline (χ2 = 2.12, p = .14).”

Lines: 179-181

As an additional editorial criticism, we can note a very large introduction to the article, in which the approaches to conducting the study are described in too much detail, but the alleged novelty of the work is completely insufficiently substantiated. The management is too stretched, psychological concepts are described in great detail and in detail, I think it needs to be shortened, reflecting what is described in the results.

We modified and shortened the Introduction

Reviewer 3 Report

Abstract

Line 18 – Please change to something like: “A total of 8 coaches and 57 children aged between 9- to 12-year old”, something to clarify that number of 57 and to include the number of coaches.

Introduction

Line 56-58 – I consider that it important to highlight that anxiety greatly influence youth sports participation, however, stating that is a predictor by authors, without evidence, it seems out of context to me. Therefore, I suggest that the Authors withdraw this statement if no evidence are available or include references if there is evidence.

Line 70 – Please put the comma in the correct place

Line 122 – Please consider changing to “In the present study, the authors used…”. Nevertheless, I would suggest to only expose that tool, explaining its possibilities and importance, removing the idea of already defining the methodology of the study

Material and Methods

Line 172 – please include a reference.

Did the coaches perform the same tasks in all experimental groups?

Discussion

The discussion seems clear and supported by the literature.

Author Response

Dear reviewer, we greatly appreciate your work. Thank you for your effort and valuable comments, which help us grow in our work and develop our abilities.

Point raised by referee

Response by author

Location in text:

Page and paragraph reference

Specific comments

Abstract

Line 18 – Please change to something like: “A total of 8 coaches and 57 children aged between 9- to 12-year old”, something to clarify that number of 57 and to include the number of coaches.

Introduction

Line 56-58 – I consider that it important to highlight that anxiety greatly influence youth sports participation, however, stating that is a predictor by authors, without evidence, it seems out of context to me. Therefore, I suggest that the Authors withdraw this statement if no evidence are available or include references if there is evidence.

Line 70 – Please put the comma in the correct place

Line 122 – Please consider changing to “In the present study, the authors used…”. Nevertheless, I would suggest to only expose that tool, explaining its possibilities and importance, removing the idea of already defining the methodology of the study

Material and Methods

Line 172 – please include a reference.

Did the coaches perform the same tasks in all experimental groups?

Discussion

The discussion seems clear and supported by the literature.

The modification is done

We added the reference

There should not be comma.

Done

Done

Done

Yes, activities were the same. We wrote in the Material and Method section (Design and Procedure subchapter): “During each workshop, the coaches were taught practical applications and were provided with activities and a timetable to implement the i7W model. The coaches were also provided with a template to record the activities they implemented”.

Line 18

Line 58

Lines: 202-204

Round 2

Reviewer 1 Report

The authors correctly made the suggested changes and improved the manuscript with suggestions from the reviewers.

Reviewer 2 Report

After the authors have made significant edits, the article can be recommended for publication